# The Interplay between Microbiota and Chemotherapy-Derived Metabolites in Breast Cancer

**DOI:** 10.3390/metabo13060703

**Published:** 2023-05-28

**Authors:** Julio Plaza-Diaz, Ana Isabel Álvarez-Mercado

**Affiliations:** 1Department of Biochemistry and Molecular Biology II, School of Pharmacy, University of Granada, 18071 Granada, Spain; jrplaza@ugr.es; 2Instituto de Investigación Biosanitaria ibs.GRANADA, Complejo Hospitalario Universitario de Granada, 18014 Granada, Spain; 3Children’s Hospital of Eastern Ontario Research Institute, Ottawa, ON K1H 8L1, Canada; 4Institute of Nutrition and Food Technology, Biomedical Research Center, University of Granada, 18016 Armilla, Spain

**Keywords:** breast cancer, microbiota, metabolites, chemotherapy, host metabolism, diet

## Abstract

The most common cancer in women is breast cancer, which is also the second leading cause of death in this group. It is, however, important to note that some women will develop or will not develop breast cancer regardless of whether certain known risk factors are present. On the other hand, certain compounds are produced by bacteria in the gut, such as short-chain fatty acids, secondary bile acids, and other metabolites that may be linked to breast cancer development and mediate the chemotherapy response. Modeling the microbiota through dietary intervention and identifying metabolites directly associated with breast cancer and its complications may be useful to identify actionable targets and improve the effect of antiangiogenic therapies. Metabolomics is therefore a complementary approach to metagenomics for this purpose. As a result of the combination of both techniques, a better understanding of molecular biology and oncogenesis can be obtained. This article reviews recent literature about the influence of bacterial metabolites and chemotherapy metabolites in breast cancer patients, as well as the influence of diet.

## 1. Breast Cancer: An Overview

Breast cancer (BC) is the most common cancer in women and the second leading cause of death in females. It is necessary to mention that some women will get BC even without a known risk factor (Figure 1), and, on the contrary, the presence of a risk factor does not necessarily result in BC. Moreover, not all risk factors have the same effect. In fact, most women have some risk factors, but most of them will not develop BC [1].

BC is mainly categorized and systemically managed based on molecular subtypes. Histopathology is the definitive method for diagnosing BC [2]. In compliance with Giuliano et al. [3], the size of the tumor, the number of nodes, and the occurrence of metastases are all considered in the staging system. Invasive carcinoma (70–75%) and lobular carcinoma (12–15%) are the two most common types of BC [4]. 

BC treatments are generally focused on curing the disease, improving disease-free survival and overall survival, and improving quality of life. Different types of modalities can be associated with this condition, including local, systemic, and supportive treatments.

It is possible to divide local treatments into surgery and radiotherapy and systemic therapies into chemotherapy, immunotherapy, and targeted and hormone therapy [4,5]. Treatment must be selected based on the size and location of the tumor, the lymph node commitment, histopathology, the molecular subtype, and the presence of metastases. The condition of the patient’s health, age, hormonal status, and preferences should also be taken into consideration [4,6].

Surgical removal of the tumor and, in some cases, radiotherapy of the axillary lymph nodes is recommended in patients with non-metastatic tumors. The majority of patients who undergo local surgery for breast conservation require radiation therapy. In addition, neoadjuvant and adjuvant systemic therapies have been utilized in conjunction with surgery [7,8]. Moreover, approximately 70% of patients are positive for the estrogen- or progesterone-receptor-positive and the human epidermal growth factor 2 (ERBB2, formerly known as HER2) negative subtype. These patients are primarily treated with endocrine therapy; some may also receive chemotherapy. Patients who have tumors that are positive for ERBB2 (15–20% of BC cases) are treated with a combination of ERBB2-targeted therapy and chemotherapy. Luminal A and B BCs should be treated with alkylator or taxane-based chemotherapy regimens combined with anthracyclines. HER2+ tumors in stages 2 or 3 should receive chemotherapy that contains anthracyclines, alkylators, and taxanes in addition to trastuzumab or pertuzumab, and those at stage 1 should receive chemotherapy that contains paclitaxel and trastuzumab [7]. 

Triple-negative tumors (those that lack estrogen receptors, progesterone receptors, and ERBB2 molecular markers) constitute 15% of BC cases and are generally treated with chemotherapy [8,9], although neither hormone therapies nor specific targets are available for triple-negative tumors. As a result, treatment options are more limited. As a result, chemotherapy is always used, even in cases where there are no metastases to distant organs or lymph nodes. Women with triple-negative BC are most commonly treated with doxorubicin combined with cyclophosphamide, doxorubicin combined with cyclophosphamide and paclitaxel, and docetaxel combined with cyclophosphamide when doxorubicin has recurred and has become resistant [8,10,11]. Besides chemotherapy, other treatment options are available for triple-negative breast tumors with metastases, including immunotherapy, Poly (ADP-ribose) Polymerases inhibitors, and cisplatin or carboplatin [6]. As monotherapy, taxanes (paclitaxel and docetaxel), platinum compounds (cisplatin), or anthracyclines (doxorubicin) are used in patients with triple-negative tumors and metastases at the time of diagnosis [10,11,12]. PARP1 inhibition is used when there is no or reduced activity of BRCA1 in the tumor, antibody treatments are used when the tumor overexpresses the Epidermal Growth Factor Receptor (EGFR), c-KIT tyrosine kinase inhibitors are used when it overexpresses c-KIT, and multikinase inhibitors are used when EGFR is overexpressed [7,11,13]. Patients with metastatic BC are treated similarly to those with non-metastatic BC; however, palliative treatment is used to prolong the patient’s life [8] (Table 1).

Considering the above, it is also worth remarking that research indicates that the composition of the gut microbiota can influence the efficacy and side effects of cancer immunotherapy. However, not much is known about the importance of the interaction between microbiota-induced and drug metabolites.

The main objective of this study is to highlight the importance of some metabolites produced by or in response to chemotherapy treatment. Additionally, this article reviews the literature about the influence of bacterial metabolites and chemotherapy metabolites in BC patients, as well as the influence of diet.

## 2. Chemotherapy as a Treatment for BC and the Importance of Metabolites in the Patient’s Response to Treatment

Under dynamic conditions, cancer cells adjust their fuel utilization to support their requirements for cellular function, proliferation, and survival [14]. There is no doubt that metabolic reprogramming is a hallmark of cancer, and several metabolic enzymes have been identified as potential drug targets for the treatment of cancer [15]. A uniform metabolic signature specific to a tumor has yet to be identified. Tumors display distinct metabolic programs, which makes targeting metabolism in different types of tumors challenging. 

Tumor microenvironments are characterized by a variety of metabolite compositions, which are regulated at various levels. There is a direct relationship between local nutrient availability and tumor cell metabolism, as well as metabolic cross-talk between tumor cells, infiltrating immune cells, and supporting stromal cells. The consumption and secretion of metabolites by all of these cell types affect the tumor microenvironment and influence the conditions that result [16]. However, the anatomical location of the tumor also affects the metabolic microenvironment. As a result of tissue structure, levels of perfusion, and function, metabolic heterogeneity can be found in different organs, as well as in sub-local organs and tumors [16]. Furthermore, changes in metabolite availability may also result from changes in systemic metabolism, such as changes in dietary intake, changes in the functions of metabolic organs, or the presence of metabolic syndromes. All of these levels contribute to the heterogeneity of the tumor microenvironment, and each has its therapeutic potential [16].

In addition to targeting tumor cells, many drugs also cause collateral damage that compromises immune cell function. Even though glycolytic inhibitors affect tumor cell growth, they also abrogate the immune surveillance of cancer cells [17,18]. Accordingly, there are ongoing phase I clinical trials for the inhibitor of the MCT1/2 lactate transporter (AZD3965) in both solid tumors and large B-cell lymphomas [19]. The inhibition of this transporter, however, also adversely affects the proliferation of T cells. Nucleotide synthesis requires one-carbon units for biosynthetic processes [20]. Folate metabolism plays an important role in this process. Antifolates, such as methotrexate, are widely used in the treatment of cancer. The proliferation and survival of CD4+ and CD8+ T cells both require one-carbon metabolism, suggesting that antifolate therapy may be detrimental to the immune system [21,22].

Many types of cancer are resistant to conventional chemotherapy [23]. Resistance to chemotherapy is characterized by a drug’s low ability to produce a beneficial response in treatment and is also one of the main causes of chemotherapy failure [23,24,25]. Several mechanisms are associated with tumor chemoresistance, including interactions between cancer cells and the tumor microenvironment.

The number of studies investigating how chemotherapy affects BC metabolism is still limited [26,27,28,29]. NMR spectroscopy of 1H high-resolution magic angle spinning is used to study the effects of chemotherapy on the metabolic profile of ER-positive human BC cells MCF-7 [30]. Using this technique, it was confirmed that doxorubicin, cisplatin, and tamoxifen significantly altered the metabolic profile of MCF-7 cells [30].

A study was conducted in 1992 on metabolites of limonene (an anticancer drug). The two major circulating metabolites of limonene in rats, perillic acid and dihydroperillic acid, inhibit protein isoprenylation more potently than limonene. Perillic acid also inhibits cell growth more strongly [31].

A significant difference was found between the concentrations of four metabolites, three from nuclear magnetic resonance spectroscopy (threonine, isoleucine, and glutamine) and one from liquid chromatography-mass spectrometry (linolenic acid) when comparing chemotherapy response in BC. Based on a prediction model that combined nuclear magnetic resonance spectroscopy with mass-spectrometry-derived metabolites, 80% of patients whose tumors did not respond to chemotherapy were correctly identified [32].

In cancer biology, LAT1 is considered to be particularly important and is more abundant in cancer cells than in normal cells. LAT1 belongs to the L-type amino acid transporter family, which consists of four members (LAT 1–4) [33,34]. There is evidence that ER-positive breast carcinoma cells require leucine for proliferation, which is transported by LAT1 and, increasingly, by binding to the scaffold protein LLGL2 [35]. LAT1 promotes amino acid uptake, which contributes to energy production by providing amino acids to the TCA cycle. In vitro analysis revealed that LAT1 was involved in the amino acid uptake process. Despite decreased glucose metabolism, oncometabolites and branched-chain amino acids also played a significant role in energy production and drug resistance in MCF-7 cells treated with chemotherapeutic agents [36]. 

During chemotherapy, the tumor microenvironment can be altered by heterogeneity in cancer cells, cancer stem cells, macrophage-associated macrophages, and immune cells, thereby leading to the development of chemoresistance. Many factors can contribute to chemotherapy resistance, including intrinsic factors such as tumor heterogeneity, cancer stem cells, and epigenetics, as well as extrinsic/acquired factors such as pH, hypoxia, paracrine signaling, and other tumor cells [25,37]. BC cell resistance mechanisms include absorption, transportation, and efflux of drugs through the cell membrane, transporter proteins, cancer-associated genes, DNA repair, cancer stem cells, the tumor microenvironment, and epithelial-mesenchymal transitions [25,38]. Figure 2 summarizes the main causes of chemotherapy resistance in BC.

## 3. Breast Cancer and the Microbiota

Human-associated microbiota is composed of communities of bacteria, fungi, protists, Archaea and viruses that inhabit the human body [39]. The microbiota is essential to maintaining the homeostasis of the organism. It provides benefits by modulating immunity and signal transduction, as well as affecting many physiological and pathological processes in the human body, such as drug metabolism and vitamin synthesis. An unbalanced microbiota, referred to as dysbiosis, is related to many different diseases causing chronic inflammation and cancer [40].

The composition of gut microbiota is mediated by a variety of factors, such as host characteristics, environmental pollutants, geographical location, bacterial infections, antibiotic treatment, lifestyle, surgical procedures, and age. Although once established, the composition of the gut microbiota is relatively stable throughout adult life [41], it can be altered, as diet as one of the main modulators of gut microbiota. In this sense, diet would be two sides of the same coin, as an inadequate diet promotes obesity and gut dysbiosis (both risk factors associated with cancer), but at the same time, adherence to healthy dietary patterns is known to be a useful strategy for the treatment of overweight and obesity and the maintenance of a balanced microbiota [42,43].

Several studies have examined the role of the gastrointestinal microbiota in carcinogenesis [44,45,46,47,48]. Indeed, the relationship between cancer and the microbiota is not surprising considering that the latter may also induce detrimental effects on the host [40]. In addition, the ability of the microbiota to induce chronic inflammation and the potential of certain bacterial species to damage the DNA double helix are mechanisms that are applicable to cancer occurrence [49,50]. Another very critical risk factor for developing and promoting cancer is the microbiota’s functionality. Growing evidence also indicates that the human gut microbiota interacts with xenobiotics, such as endocrine disruptors. These can disrupt microbial communities and decrease signaling pathways that compromise their functions [51]. The microbiota might also promote drug efficacy, ameliorate the harmful effects of toxic drugs, and prevent cancer [52]. Other studies have suggested its suitability for diagnosing, predicting risk and course, and preventing this disease [53].

Concerning breast tissue microbiota and their relation to BC, numerous bacterial species present on the skin may access the milk ducts. Various studies suggest that intestinal bacteria can reach the breast tissue, promoting tumor genesis. 

As in the digestive tract, a specific microbiome is beneficial in maintaining a healthy mammary microenvironment. In fact, there is evidence that is beginning to corroborate this hypothesis. For example, antibiotic treatment of mice with breast tumors resulted in tumor growth [54,55]. Other authors have described the cancerous breast as having a higher relative abundance of *Bacillus*, *Staphylococcus*, and *Enterobacteriaceae*, and the two species isolated from the affected breast, *Escherichia coli* and *Staphylococcus epidermidis*, can cut the DNA of HeLa [49,50]. It has been also suggested that changes in the microbiota may promote tumor development and be involved in its aggressiveness. BC patients have lower microbiota alpha diversity, a lower number of species, and a different composition of microbiota compared to healthy controls [56]. Moreover, it has been shown that the histological grade of BC impacts the tumor’s microbial profile [57]. When the tumor grade increased, the compositional abundance of the *Bacteriodaceae* family was reduced. Further, an increase in the genus *Agrococcus* has been observed as tumor malignancy increases [57].

On the other hand, the chemotherapy response has also been linked to cross-talk between the gut microbiota and the anticancer agents in many studies [58].

The gut microbiota is capable of modulating the activity, efficacy, and toxicity of chemotherapy/immunotherapy agents [40,59]. On the contrary, several studies have shown that chemotherapeutic drugs negatively affect the intestinal microbiota, induce dysbiosis, and influence metabolism. In fact, cancer is highly affected by the immunological aspects of the microbiome [60,61,62,63], alters the intestinal microbiota, and, in turn, also affects the effectiveness and toxicity of chemotherapy [64]. 

Research into the human microbiomes of various types of cancer is increasing the demand for metabolites in cancer therapy and immunosurveillance [65].

Chemotherapy induces intestinal microbiota dysbiosis and intestinal mucositis, a painful and debilitating condition affecting up to 40–100% of patients undergoing this treatment. During chemotherapy, cytotoxic agents such as antimetabolites, alkylating agents, platinum complexes, topoisomerase inhibitors, or antibiotics are effective at destroying cancer cells. Nevertheless, these drugs indiscriminately affect some healthy cells, promoting undesirable side effects that reduce the patient’s quality of life and increase healthcare costs [66]. The microbiota can improve chemotherapy efficacy and toxicity and even serve as a complementary treatment for cancer. Thus, a deeper understanding not only of the composition but also of the functionality of the microbiota (the latter through its metabolites), as well as its interaction with the different treatment drugs administered, seems a hopeful tool for cancer management. This knowledge would facilitate the development of therapies that are effective against the disease and capable of reducing undesirable side effects. In this line, animal studies show that mice whose microbiota has been modulated based on diets rich in prebiotics show enhanced effects of chemotherapy and radiotherapy [67]. In addition, through the modulation of the intestinal microbiota by antibiotics, the response to chemotherapy can be altered in a mouse model of triple-negative BC [60].

According to several studies, the gut microbiota facilitates and abrogates neoadjuvant therapy efficacy. By shortening the intestinal villi and causing the intestinal barrier to become impermeable, cyclophosphamide (the most commonly used chemotherapy for BC) facilitates the translocation of commensal bacteria to secondary lymphoid organs, such as *Enterococcus hirae* and *Barnesiella intestinihominis* [68] (Table 2). As soon as *E. hirae* enters the lymphoid organs, it contributes to the accumulation of type 17 and type 1 T helper cells, and *B. intestinihominis* increases the levels of polyfunctional cytotoxic CD8+ T cells within the body [69,70]. Based on shotgun metagenomic sequencing and metabolomic analysis of patients who respond to treatment, the microbiota-derived metabolite indole-3-acetic acid is enriched in patients who are responding to treatment. In humanized gnotobiotic mouse models of pancreatic ductal adenocarcinoma, both fecal microbiota transplantation and short-term dietary manipulation of tryptophan increase the efficacy of chemotherapy [71]. Using a combination of loss-of-function and gain-of-function experiments, neutrophil-derived myeloperoxidase is used to assess the efficacy of indole-3-acetic acid and chemotherapy. In combination with chemotherapy, myeloperoxidase oxidizes indole-3-acetic acid, which causes the enzymes glutathione peroxidase 3 and glutathione peroxidase 7 to be degraded. As a result of these factors, cancer cells accumulate reactive oxygen species and autophagy is downregulated, which compromises their metabolic fitness, and ultimately, their proliferative ability [71]. The levels of indole-3-acetic acid and the efficacy of therapy in two independent cohorts of patients with pancreatic ductal adenocarcinoma were significantly correlated in humans [71].

A study was conducted to examine the metabolic profiles of feces from eight patients following chemotherapy treatment cycles. In contrast to patients before treatment, amino acids were upregulated, while lactate and fumaric acid were downregulated. A significant difference was also observed between the groups studied in terms of SCFAs. Chemotherapy treatment has a significant impact on the fecal metabolomic profile of patients with BC [72].

Another important emerging factor is the tumor-resident intracellular microbiota, which has been documented for a wide variety of cancer types and has unclear biological functions [73].

Microbes have also been shown to be integral components of tumor tissue in a wide variety of cancer types beyond colorectal cancer in recent years, including pancreatic cancer, lung cancer, breast cancer, and others, which were originally thought to be sterile [50,74,75,76,77,78]. It has been shown in clinical studies [79] that certain features of tissue-resident microbiota are associated with cancer risks [79], pathological types [76,80,81,82,83], and cancer prognosis [77], as well as treatment responses [76,84,85]. 

In spite of this, tissue-resident microbiota samples typically consist of very low biomass, are often contaminated by the host, and are potentially subject to environmental noise, which often obscures the findings [86,87,88,89,90,91]. In immunodeficient mice, studies using patient-derived xenograft models suggest that the intratumor microbiota can persist as the tumor progresses [92]. In addition to impairing tumor chemosensitivities [85,93], the administration of exogenous bacteria promotes tumor progression [94].

A murine spontaneous breast-tumor model, MMTV-PyMT, was used to explore the functional significance of these intratumor bacteria. The depletion of intratumor bacteria significantly reduces lung metastasis without affecting the growth of the primary tumor. Through reorganization of the actin cytoskeleton, the intratumor bacteria carried by circulating tumor cells promote host-cell survival during metastatic colonization. Furthermore, in two murine cancer models with significantly different metastasis potential levels, the intratumor administration of selected bacteria strains from tumor-resident microbiota increased metastasis. This study suggests that tumor-resident microbiota play a significant role in promoting cancer metastasis, despite their low biomass, and that interventions may be worth exploring as part of the advancement of oncology care [73]. 

Given the fact that bacterial metabolites can interfere with biological pathways, it is worth thinking about metabolites produced by the BC patient’s microbiome that lead to changes in their metabolism. An overview of the relationship between microbes present in breast tissue and the risk of BC can be found in Figure 3. 

**Table 2 metabolites-13-00703-t002:** Breast cancer and microbiota studies.

Author	Microbiota Changes in BC
Urbaniak et al. (2016) [50]	The relative abundance of *Bacillus*, *Enterobacteriaceae*, and *Staphylococcus* was higher in women with BC.
Goedert et al. (2015) [56]	A statistically significant difference was observed between case patients and controls in terms of microbiota composition and α-diversity.
Meng et al. (2018) [57]	In malignant tissue, enriched microbial included *Propionicimonas* and *Micrococcaceae, Caulobacteraceae, Rhodobacteraceae, Nocardioidaceae, and Methylobacteriaceae*, which appeared to be ethnospecific.
Bawaneh et al. (2022) [60]	The results of a metagenomic analysis indicated that doxorubicin increased the proportional abundance of *Akkermansia muciniphila*.
Viaud et al. (2013) [68]	There was a reduction in pT(H)17 responses in tumor-bearing mice treated with antibiotics to kill Gram-positive bacteria, and their tumors were resistant to cyclophosphamide treatment.

Abbreviations: BC, breast cancer.

## 4. Breast Cancer and Bacterial Metabolites

The first reports of the involvement of bacterial metabolites in cancer were published in 1970 [95] about aflatoxins, a group of extremely potent carcinogens produced by *Aspergillus flavus*. Similarly, *Fusarium* species produce potent carcinogenic mycotoxins, as do *Streptomyces hepaticus* [96].

Some recent studies about bacterial metabolites with a potential relationship to BC are summarized below (Table 3).

### 4.1. Cadaverine

Cadaverine is produced by the decarboxylation of lysine that is performed by lysine decarboxylase enzymes. Human cells code and express numerous bacterial species of the human microbiome. Additionally, they express lysine decarboxylase either in a constant or in an inducible way. In the early stages of BC, the concentration of bacterial diamine cadaverine decreases, which leads to a decrease in the production of other anticancer bacterial metabolites [97]. 

### 4.2. Lithocholic Acid

Lithocholic acid is a monohydroxy bile acid produced by the gut microbiota [98]. Reported results were a reduction in the proliferation of cancer cells (by 10–20%) and a reduction in vascular endothelial growth factor (VEGF) production (by 37%) when the authors used concentrations that correspond to the tissue reference concentrations of lithocholic acid. Additionally, through induction of mesenchymal-to-epithelial transition, increased antitumor immunity, and the oxidative phosphorylation (OXPHOS) system, the authors were able to reduce the aggressiveness of primary tumors and their metastatic potential. In part, these effects were attributed to the activation of the Takeda G-protein-coupled bile acid receptor (TGR)-5 by lithocholic acid [99]. 

In early BC patients, compared to control women, the serum levels of lithocholic acid, the ratio of chenodeoxycholic acid to lithocholic acid, and the amount of 7α/β-hydroxysteroid dehydroxylase gene (a key enzyme in lithocholic acid generation) in fecal DNA were significantly reduced. Therefore, there may be a decrease in microbial production of lithocholic acid in early BC patients [100]. As the BC stage advanced, the expression of key components of the lithocholic acid-elicited cytostatic pathway (inducible nitric oxide synthase (iNOS) and 4-hydroxynonenal (HNE)) gradually decreased. In tumors, there was a negative correlation between lipid peroxidation and the mitotic index. Overexpression of iNOS, neuronal nitric oxide synthase (nNOS), Kelch-like ECH-associated protein 1 (KEAP1), NADPH Oxidase 4 (NOX4), and TGR5 or downregulation of nuclear factor erythroid 2-related factor 2 (NRF2) was associated with improved survival in BC patients, except for those with triple-negative disease. 

In sum, lithocholic acid, a metabolite of the gut microbiome, induces oxidative stress, which slows down the proliferation of BC cells when taken together. As BC progresses, the lithocholic acid-oxidative stress protective pathway is lost, and this loss correlates with a poor prognosis [101].

### 4.3. Indolepropionic Acid

Indolepropionic acid is a gut-bacteria-produced tryptophan metabolite. In the study reported by Sári et al. [102], some receptors in the body responded to indolepropionic acid, including the aryl hydrocarbon receptor (AHR) and the pregnane X receptor (PXR). BC patients with higher PXR and AHR expression had better survival rates, supporting the role of indolepropionic acid-induced pathways in cytostasis. Moreover, the level of activation of the AHR and the level of PXR expression were inversely related to the level of proliferation of cancer cells and the stage and grade of cancer. When women are newly diagnosed with BC, especially at stage 0, their fecal microbiome’s ability to produce indolepropionic acid is suppressed. In humans, the biosynthesis of indole by bacteria was correlated with the infiltration of lymphocytes into tumors [102].

### 4.4. Succinate

Succinate is produced in large amounts during the bacterial fermentation of dietary fiber [103]. The study by Anil Yadav showed poor relapse-free survival in ER+ BC patients with low levels of succinate dehydrogenase assembly factor 2 (SDHAF2) [104]. Moreover, by suppressing succinate dehydrogenase in tumor cells, BC-associated macrophages promoted tumorigenesis [105]. On the other hand, the accumulation of succinate occurred in hypoxia promoted BC treatment resistance, poorer patient outcomes and lower survival rates [106].

### 4.5. Short-Chain Fatty Acids (SCFAs) 

Microbes in the gut produce SCFAs, acetate and butyrate, secondary bile acids, polyamines, and vitamins that may have an impact on the development of cancer [107].

A growing body of evidence suggests that SCFAs, particularly butyrate, and propionate, can enhance chemotherapeutic agents’ effectiveness by increasing tumor sensitivity or enhancing antitumor immune responses [108]. Research indicates that decreased abundances of SCFA-producing taxa (*Coprococcus, Dorea* and uncultured *Ruminococcus)* are associated with a lower efficacy of neoadjuvant chemotherapy (cyclophosphamide, anthracycline, taxol, or herceptin) in breast cancer patients and are associated with a lower number of intratumoral CD4+ and CD8+ cells as well as peripheral CD4+ T cells [108,109]. 

Among survivors of BC who showed lower microbial diversity, increased *Bacteroidetes*, and decreased *Firmicutes*, the first evidence to suggest an association between the gastrointestinal microbiota composition and fear of cancer recurrence was reported [110]. As a result of a pilot study examining psychosocial factors and gastrointestinal microbiota composition in 12 BC survivors, fatigue has been associated with changes in the abundance of SCFA-producing *Faecalibacterium* and *Prevotella*, as well as anxiety, with changes in the abundance of *Coprococcus* and *Bacteroides* [111].

### 4.6. P-Cresol

P-cresol is the product of the breakdown of tyrosine by intestinal bacteria [112] and has been to cancer cells progression.

### 4.7. Tryptophan Metabolites

An immunohistochemistry analysis of 203 BC samples showed that none exhibited negative staining for indoleamine 2,3-dioxygenase [113]. Indoleamine 2,3-dioxygenase expression was higher in large, node-positive, and ER+ tumors, while indoleamine 2,3-dioxygenase was lower in less vascularized tumors [113]. In this study, there was a greater infiltration of CD8+ and CD11b+ cells in low-indoleamine 2,3-dioxygenase tumors, whereas T_reg_ infiltration was not associated with indoleamine 2,3-dioxygenase expression [113]. According to another study, a higher level of stromal indoleamine 2,3-dioxygenase was associated with a worse disease-free and metastasis-free survival in patients with BC [114].

Indoleamine 2,3-dioxygenase activity is commonly assessed through plasmatic kynurenine/tryptophan ratios, although they may not be a full reflection of tumor tryptophan metabolism. In patients with BC, Lyon and colleagues found a higher plasmatic kynurenine/tryptophan ratio, but there was no statistically significant difference in plasmatic levels of the two molecules [115]. It was also observed that kynurenine and tryptophan levels in the plasma of BC patients were significantly lower than those in healthy controls, mainly in patients with ER- tumors and at an advanced stage of the disease [116]. As compared to ER+ cancers, Tang and colleagues observed higher plasma kynurenine levels in ER- cancers [117].

According to the above studies, indoleamine 2,3 dioxygenase 1 is elevated in tumor cells to suppress immune surveillance and promote tumor growth. This suggests that indoleamine 2,3 dioxygenase 1 might be used as a therapeutic target for the treatment of BC. Several studies have shown either an increase in indoleamine 2,3 dioxygenase 1 activity [118,119,120] or no difference [121,122] after receiving chemotherapy, such as paclitaxel, Mohs paste, or surgery. Although limited information is available on the patient cohort, potential explanations for these different observations could be due to the proportion of patients with BC subtypes and/or the percentage of patients with cancers expressing indoleamine 2,3 dioxygenase 1. In addition to indoleamine 2,3 dioxygenase 1, other rate-limiting enzymes have been investigated, such as tryptophan 2,3 dioxygenase and the downstream enzyme of the kynurenine pathway, kynurenine monooxygenase. There has been a positive correlation between tryptophan 2,3 dioxygenase levels and poorer overall survival, increased disease grade, and invasion/migration capability in five studies [116].

Furthermore, indoleamine 2,3 dioxygenase 1 is often detected concurrently with tryptophan 2,3 dioxygenase expression [123,124,125,126,127]. Additionally, tryptophan 2,3 dioxygenase expression correlated strongly with the expression of aryl hydrocarbon receptors and has been associated with enhanced tumor cell migration [123,126,127,128]. In the triple-negative BC subtype, kynurenine monooxygenase was found to be elevated, and it was associated with poorer survival and malignant characteristics such as node positivity. There is a high frequency of kynurenine monooxygenase amplification in invasive BC compared to all other human cancers [129,130,131,132].

Among the small number of enzymes capable of catabolizing tryptophan, Indoleamine 2,3 dioxygenase 1 has emerged as an attractive pharmacological target due to its well-characterized structure and high affinity for tryptophan. The fact that it is the rate-limiting enzyme means that the inhibition of its activity may cause the whole kynurenine pathway to be blocked or reduced, and the degree of inhibition in the blood can be measured indirectly. Since most human cancers, including BC, overexpress indoleamine 2,3 dioxygenase 1, indoleamine 2,3 dioxygenase inhibitors could be used clinically in enhancing antitumor immunity. It has been demonstrated in in vivo and in vivo studies that combination treatment of indoleamine 2,3 dioxygenase 1 inhibition and chemotherapy drug(s) limits tumor growth. ClinicalTrials.gov has reported 97 different clinical trials examining combination therapy in cancer patients [133]. A total of six clinical trials involving patients with BC have been conducted; the different inhibitors of indoleamine 2,3 dioxygenase 1 examined were epacadostat, indoximod, and navoximod/GDC-0919 [134].

Mechanistically, SCFAs may play a significant role in gut–brain communication through their effects on blood–brain barrier permeability, microglial activity, neuronal function, and neuroinflammation [135], all of which have been reported following cancer treatment [136,137]. Succinate plays an active role in immunity and cancer through the stabilization of hypoxia-inducible factor (HIF)-1 [138]. Indolepropionic acid-mediated downregulation of NRF2 and upregulation of iNOS contribute to increased oxidative/nitrosative stress. An increase in oxidative/nitrosative stress reduces the stemness of cancer cells and leads to cytostasis [102]. Intestinal bacteria break down tyrosine into p-cresol [112], which contributes to the progression of cancer cells. The effect of lithocholic acid on BC was studied through oxidative stress and apoptosis [66,139,140]. During T-regulatory cell differentiation, indoleamine 2,3-dioxygenase converts tryptophan into kynurenine, an immunosuppressive metabolite. Many types of cancer, including BC, express Indoleamine 2,3-dioxygenase [141]. A decline in bacterial diamine cadaverine concentration in the early stages of BC leads to a decrease in bacterial metabolites that are anticancer in nature [97] (Table 3).

**Table 3 metabolites-13-00703-t003:** Breast cancer and bacterial metabolites.

Author	Metabolites	Main Effects in BC
Mikó et al. (2018) [100]	Lithocholic acid	Proliferation and the aggressiveness of cells are reduced.
Kovács et al. (2019) [101]	Lithocholic acid	Lithocholic acid induces oxidative stress, which inhibits the proliferation of procancer cells.
Sári et al. (2020) [102]	Indolepropionic acid	BC inhibits the production of indolepropionic acid, a cytostatic metabolite produced by bacteria.
Ravnik et al. (2021) [99]	Cadaverine, succinate, p-cresol	Cadaverine, succinate, p-cresol, and their derivatives may be useful in the diagnosis of BC.
Yadav et al. (2023) [104]	Succinate	Modulation of succinate metabolism may contribute to restoring sensitivity to fulvestrant and tamoxifen resistance.
Gomez et al. (2020) [105]	Succinate	Angiogenesis and immunosuppression are promoted by tumor-associated macrophages by suppressing succinate dehydrogenase.
Kang et al. (2017) [106]	Succinate	BC’s aggressiveness is driven by succinate.
Soliman et al. (2013) [113]	Tryptophan	There was a higher expression of Indoleamine 2,3 dioxygenase in ER+ tumors compared to ER− tumors. In patients with higher neoangiogenesis, Indoleamine 2,3 dioxygenase levels were lower. Patients with high levels of indoleamine 2,3 dioxygenase expression had a better overall survival.

Abbreviations: BC, breast cancer.

In addition to being an emerging hallmark of cancer, metabolic reprogramming is also a key regulator of cancer cells’ ability to adapt to their microenvironment. The use of metabolic imaging for the diagnosis of BC has been widely accepted. Even though metabolic plasticity is a significant contributor to therapy resistance in BC, limited implications have been explored for monitoring treatment response [142].

Finally, the analysis of serum levels of inflammation-related markers and metabolites in cohorts may also provide actionable targets to enhance the effect of antiangiogenic therapies currently being used in many types of cancer, including BC [143,144,145].

## 5. Modulation of the Microbiota through Diet as a Potential Adjuvant/Complementary Treatment for Breast Cancer

Metabolic alterations (e.g., Warburg effect, glutamine addiction, or increased serine biosynthesis) have been shown to play a role in the development and maintenance of various types of cancer [14,146,147,148,149]. Because metabolism is intimately associated with the signaling pathways that control cell death [150], it is becoming increasingly evident that metabolic adaptations may also contribute to both chemotherapeutic and targeted resistance [151,152,153]. During and after chemotherapy, changes in resting energy expenditure are observed, accompanied by changes in body composition, revealing a U-shaped curve. According to this hypothesis, changes in fat-free mass are associated with variations in resting energy expenditure as a consequence of the catabolic effects of the disease and the administered medication. As a result of the Harris–Benedict equation, resting energy expenditure is often underestimated, illustrating the importance of assessing a patient’s energy requirements and providing nutritional support [154].

As mentioned, the gut microbiota is a dynamic organization that is influenced by complex interactions between different factors during cancer therapy. These factors include host immunity, chemotherapeutics, concurrent medications, the environment, and diet [59]. 

Among the tools for manipulating the gut microbiota, dietary modifications are one of the most studied. In this regard, there is a growing interest in nutrition and cancer as key factors in both quality of life and the pathophysiology of cancer [155].

Regarding the triad of cancer–diet–microbial metabolites, it is well known, for example, that protein residues and fat-stimulated bile acids are metabolized by the microbiota into inflammatory and/or carcinogenic metabolites, which increase the risk of neoplastic progression. On the contrary, SCFA, which are the main metabolites produced by the microbiota in the large intestine through the anaerobic fermentation of indigestible polysaccharides [156], can modulate the inflammatory response [157,158]. 

Concerning BC in particular, the incidence and progression of this disease are profoundly influenced by dietary and hormonal factors and by the history of pregnancy and lactation. In addition, obesity has been related to increased infiltration of immunosuppressive macrophages in BC lesions, which is also an indicator of poor prognosis for these patients [9]. On the other hand, breast microbiomes are not in direct contact with cancer. However, secondary bile acids and bacterial metabolites that are absorbed and recognized by the microbiome may have a negative impact. This is because they can infiltrate the circulation of body fluids and alter signaling pathways [9,100]. In conclusion, restoring the balance of the microbiota through dietary intervention and identifying metabolites directly related to the development of cancer and its complications would help to establish and improve treatments and define recommendations and demands from the public and private sectors to address a serious problem of world health.

### 5.1. Fecal Microbiota Transplantation

In fecal microbiota transplantation (FMT), which dates back at least 1700 years, fecal microbes are transferred from a healthy donor into the patient’s intestine in an effort to restore the balance of microbiota [159]. It has been widely studied for its ability to target and modulate the human gut microbiota, which has been implicated in the treatment of numerous diseases [160]. FMT exhibits significant advantages over other treatment modalities due to its ability to maintain microbial diversity without disrupting the natural balance of the gut microbiome. Infections caused by *Clostridioides difficile* are commonly treated with FMT [161,162]. Currently, no studies or results have been found in Medline, EMBASE, or clinicaltrials.gov on BC and FMT.

### 5.2. Pre/Probiotics

Gut microbiota play an important role in estrogen metabolism, which contributes to carcinogenesis. Estrogens have been shown to be deconjugated by beta-glucoronidase-producing bacteria. A deconjugation of estrogen promotes its re-uptake, which may influence the risk of recurrence. Bile acid conversion is also influenced by intestinal microbes. Bile acids have similar effects to hormones and can alter metabolic pathways in distant tissues by activating receptors such as farnesoid X, which has been detected in cases of invasive BC. A murine study demonstrated that *Helicobacter hepaticus* in the gut mediates the neutrophil response to breast cancer, thereby accelerating its progression [163]. As a result of chronic inflammation and epigenetic deregulation, gut bacteria can further facilitate BC development [164].

A host’s gut microbiome affects the response to chemotherapy by facilitating drug efficacy or compromising anti-cancer effects and mediating toxicity. In a recent randomized trial, probiotics containing *Bifidobacterium*, *Lactobacillus*, and *Enterococcus faecalis* significantly reduced the risk of chemotherapy-related cognitive impairment among women receiving adjuvant chemotherapy for BC [165]. Anti-HER2 therapy is associated with specific microbiota clusters (*Clostridiales, Bacteroides*). In addition, there is evidence that the microbiome of the gut influences the degree of response to immune checkpoint inhibitors CTLA4 and PD-L1/PD-1 [166]. Researchers have observed that mice with specific microbiota (e.g., *Akkermansia* and *Bifidobacterium*) respond better to anti-PD-L1 therapy and that patients with *Bacteroidetes* have a greater resistance to immune checkpoint inhibitor-induced colitis [164,167].

For 4 months, 34 BC survivors were randomly assigned either to follow a Mediterranean diet plus one sachet of probiotics per day (*Lactobacillus rhamnosus* HN001, *Bifidobacterium longum* BB536) or to follow a Mediterranean diet alone (control group, *n* = 18). Compared to an Mediterranean diet alone, probiotics were found to improve metabolic and anthropometric parameters in BC survivors with a combination of a Mediterranean diet and probiotics [168]. In another study, by regulating antitumor immune responses, *Lactobacillus acidophilus* was found to delay the development of BC [169].

### 5.3. Exercise

A BC patient undergoes various treatments within one year of being diagnosed, depending on the subtype of the tumor and its stage. Symptoms associated with each treatment may negatively impact patients’ health and quality of life. Exercise interventions can mitigate these symptoms when applied appropriately to patients’ physical and psychological conditions [170]. Various situations and exercise routines may be related to BC, to patients who are undergoing treatment, to survivors, and to patients who are receiving several medications in connection with BC. In this section, we summarize the findings of several systematic reviews.

Exercise at home and its effects on physical fitness (cardiorespiratory fitness, muscle strength, and body composition) in BC patients who are undergoing active treatment have been studied. Exercise programs conducted at home regularly are an effective strategy for improving 6 min walk tests in BC patients undergoing active treatment. In contrast, neither muscle strength nor body composition was altered [170]. The study included 13 randomized controlled trials involving 1569 breast cancer patients. Compared with control groups, groups who performed interventions that combine exercise and diet demonstrate significant improvements in cardiorespiratory fitness, muscle strength, body composition, quality of life, fatigue, anxiety, depression, and sleep. Conversely, exercise combined with supplementation does not result in an improvement over exercise alone or supplementation alone [171].

For cancer-related fatigue in women with BC, yoga, aerobic resistance, and aerobic yoga are recommended as inter- and post-treatment exercises to improve their physical resilience and quality of life in the long run [172]. 

Patients with breast cancer exhibited improved physical activity behavior for several months following exercise interventions, though the effects were small to moderate and diminished over time. After the completion of an exercise intervention, future studies should clarify how to maintain a healthy level of physical activity [173].

Oncology rehabilitation can benefit from the use of exergames, especially in BC. It is, however, necessary to conduct more rigorous studies to evaluate the effectiveness of using exergames in conjunction with conventional rehabilitation and to determine whether participants are satisfied, motivated, and adherent to the program [174].

### 5.4. Specific Nutrients, Bioactive Compounds

BC survivors are highly likely to use dietary supplements. Diindolylmethane, a compound naturally found in cruciferous vegetables, is one of the most widely used supplements. A significant amount of experimental evidence supports the bioactivity of this bioactive compound in the prevention of BC. There is a lack of well-designed human clinical trials testing its efficacy and safety. For a minimum of 12 months, women taking tamoxifen for primary or tertiary prevention of BC were randomly assigned to receive 150 mg diindolylmethane twice daily or a placebo. The results of this study demonstrate a favorable shift in estrogen metabolism and sex hormone binding globulin in the setting of breast cancer chemotherapy. In spite of this, the reduced levels of tamoxifen metabolites raise concerns about the potential interaction between diindolylmethane and tamoxifen, which requires further research. These data will contribute to informing the use of diindolylmethane as a dietary supplement for BC patients receiving tamoxifen, given the widespread and generally unsupported use of dietary supplements by BC survivors [175].

In a small group of 30 patients with breast cancer, the protective effect of 6 g per kilogram of curcumin was assessed throughout the course of radiotherapy. In the interventional group, 28.6% of patients had moist desquamation, compared to 87.5% of patients in the control group [176].

Various compounds have been shown to have different levels of action regarding proliferation, apoptosis, and metastasis in breast cancer patients, mainly in vitro. Accordingly, resveratrol has demonstrated the ability to reduce negative features by acting on ER, EFGR/PI3K, and ERK1/2 pathways. For specific cases, other phytochemicals, such as lignans or curcumin, may also be beneficial in inhibiting the HER-2 pathway [177]. A new path for future research is opened by clinical evidence derived from the use of other compounds (isoflavones). In some Asian countries, consumption of this food has been associated with a lower rate of BC [177]. 

Furthermore, a prospective Spanish trial demonstrated that the Mediterranean diet protects against the development of BC [178,179]. It is recommended that caution be exercised when interpreting these publications, as they may incur selection bias.

## 6. Further Directions

Certainly, chemotherapy, radiotherapy, and surgical intervention are still the main modalities for the treatment of BC despite continuous advances in medical therapies such as targeted therapy and immunotherapy. In addition, a large body of evidence suggests that the composition of the gut microbiota can influence the efficacy and side effects of cancer immunotherapy. However, not much is known about the influence of the interaction between microbiota-induced and drug metabolites in BC.

Between the known factors that can modify the composition of the microbiota associated with BC, dietary modifications seem to be one of the most affordable, although this assertion is somewhat controversial when viewed from the point of view of the new dietary patterns that are being imposed nowadays (e.g., the Westernized diet) as a consequence, in turn, of lifestyle changes.

Age is another known risk factor for BC that also brings with it menopause and the hormonal consequences that also predispose individuals to weight gain. As mentioned, weight gain can lead to obesity and dysbiosis of the intestinal microbiota.

Thus, and given the fact that the cell environment plays an essential role in the oncogenesis and tumor cells’ phenotypes, a deep knowledge of the microbiome and its derived metabolites has the potential to serve as a powerful biomarker of clinical response to chemotherapy. One of the recognized hallmarks of cancer cells is deregulated cellular metabolism, characterized by enhanced metabolic autonomy compared with non-transformed cells [180]. In this respect, metabolomics integrates the impact of the cell’s environment on cell biology [181]. Metabolomics analyses are fast, not too expensive, and compatible with other routine practices. In oncology, metabolomics highlights the main metabolic disturbances and the interaction of tumors. It also identifies the metabolic pathways involved in oncogenesis using the tumor cells’ metabolites profiles [182].

Additionally, microbial manipulation, through diet, exercise, prebiotics, probiotics, or microbial-derived metabolites, could promote antitumor immune responses. For this reason, more well-standardized intervention studies are needed to reveal mechanisms of microbiota homeostasis and the toxic and side effects of BC chemotherapy. Therefore, metabolomics is a complementary approach to metagenomics to gain knowledge of the impact of the extracellular environment on the tumor cell phenotype. Together, both techniques can lead to an increase in knowledge of molecular biology and understanding of oncogenesis.

However, it is not only greater knowledge of microbiological and metabolomic that is needed to develop therapeutic targets to reduce the side effects of chemotherapy and increase its efficacy. Additionally, it is imperative to consider that other factors such as the patient’s dietary pattern can play a crucial role and can undermine or jeopardize the treatment’s success.

## Figures and Tables

**Figure 1 metabolites-13-00703-f001:**
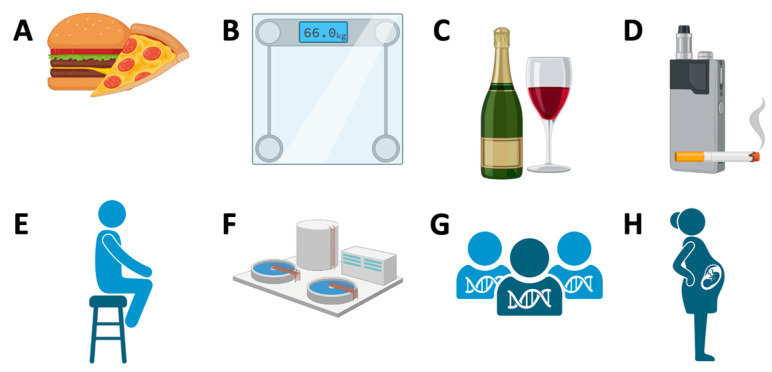
Some risk factors associated with BC: (**A**) diet; (**B**) obesity; (**C**) alcohol; (**D**) smoking; (**E**) a sedentary lifestyle; (**F**) contaminants; (**G**) genetics; (**H**) reproductive factors.

**Figure 2 metabolites-13-00703-f002:**
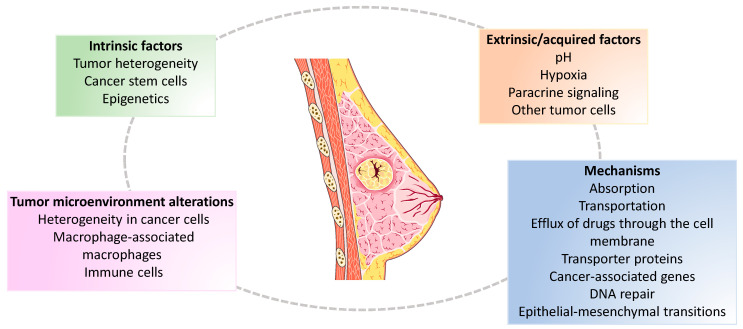
Causes of chemotherapy resistance in breast cancer.

**Figure 3 metabolites-13-00703-f003:**
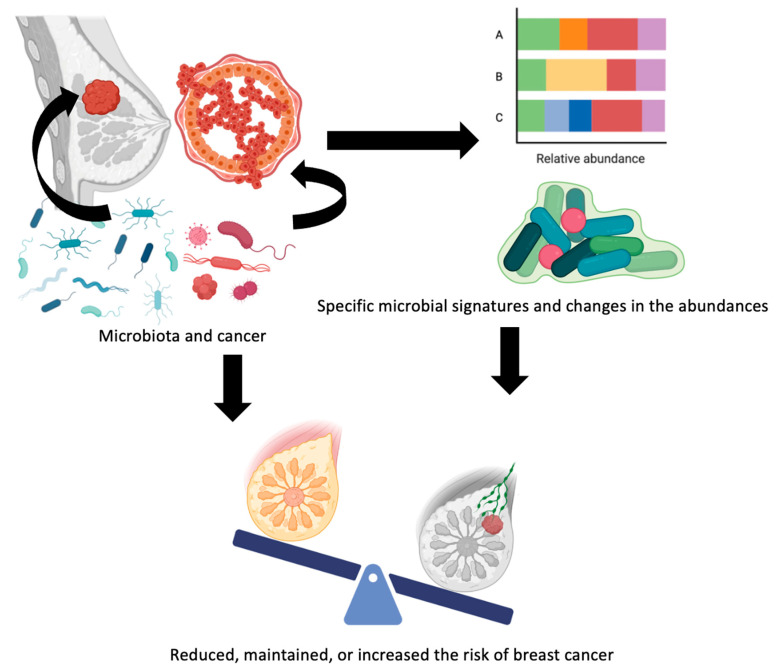
Overview of the relationship between microbes present in breast tissue and the risk of breast cancer.

**Table 1 metabolites-13-00703-t001:** Breast cancer subtypes and currently available treatments.

Subtype	Phenotype	Treatment for Non-Metastatic Tumors	Treatment for Metastatic Tumors
Luminal A	90% ER+	Endocrine treatment: Aromatase inhibitors and/or ER antagonists Chemotherapy: taxanes, anthracyclines or cyclophosphamide	First-line: endocrine therapy with aromatase inhibitors and/or ER antagonists, alone or combined with CDK4 inhibitors Second-line: mTOR inhibitors, generally in combination with endocrine therapy Others: chemotherapy with taxanes, anthracyclines or cyclophosphamide
	89% PR+		
	14% ERBB2+		
Luminal B	98% ER+	Endocrine therapy (aromatase inhibitors and/or ER antagonists) in combination with chemotherapy	
	82% PR+		
	24% ERBB2+		
ERBB2-enriched	38% ER+	Chemotherapy (with taxanes only, in patients with low tumor burden) plus trastuzumab Pertuzumab in patients with locally advanced disease, alone or combined with taxanes and trastuzumab	First-line: chemotherapy (taxanes plus trastuzumab and pertuzumab) Second-line: trastuzumab emtansine (antibody-drug conjugate) Others: chemotherapy and/or targeted therapy with RTK inhibitors
ERBB2-enriched	20% PR+		
ERBB2-enriched	72% ERBB2+		
Basal-liked	8% ER+	Chemotherapy (including carboplatin in patients with *BRCA1* mutations)	Chemotherapy (including carboplatin in patients with *BRCA1* mutations)
	7% PR+		
	7% ERBB2+		

Abbreviations: BRCA1, breast cancer 1, early onset; ER (official name ESR1), estrogen receptor 1; ERBB2; mTOR, mechanistic target of rapamycin; PR (official name PGR), progesterone receptor; RTK, receptor tyrosine kinase.

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
