# Peer review of "The Interplay between Microbiota and Chemotherapy-Derived Metabolites in Breast Cancer"

_metabolites, 2023, doi:10.3390/metabo13060703_

Round 1
Reviewer 1 Report
The review article by Julio Olaz-Diaz is very well-written. They covered adequate topics in this field. Here are my suggestions to improve the quality of the article.
1. Please prepare table for BC and microbiota.
2. Tryptophan metabolites such as I3C should be included as part of the review. Secondly, authors should consider potential mechanisms.
3. What is the impact of microbiota residing in BC should be discussed.
Author Response
Mr. Farr Liu
Assistant Editor,
Thank you for providing us with the opportunity to submit a revised version of our manuscript entitled “The interplay between microbiota and chemotherapy-derived metabolites in breast cancer” to the Metabolites journal in the Special Issue titled “Effect of Diet on Gut Microbiota and Host Metabolism”. We would like to thank the reviewers for their thoughtful comments and suggestions regarding our manuscript. All comments have been considered and incorporated into the revised manuscript. Changes to the original document are tracked and highlighted in green and blue font for the reviewer's comments, with an itemized point-by-point response to the reviewers' comments.
COMMENTS FROM REVIEWER #1
The authors would like to thank the reviewer for his/her respected comments and effort made during the review process, which are highly appreciated.
Point-to-point response:
1.- The review article by Julio Plaza-Diaz is very well-written. They covered adequate topics in this field. Here are my suggestions to improve the quality of the article. Please prepare table for BC and microbiota.
Response: Thanks to the reviewer for his/her kind comment about our manuscript. Table 2 (Breast cancer and microbiota studies) was added to the breast cancer and microbiota section (page 9, lines 312-314).
2.- Tryptophan metabolites such as I3C should be included as part of the review. Secondly, authors should consider potential mechanisms.
Response: We have added the tryptophan metabolites section, pages 11-12, lines 402-454
“4.7. Tryptophan metabolites
An immunohistochemistry analysis of 203 BC samples showed that none exhibited negative staining for indoleamine 2,3-dioxygenase [113]. Indoleamine 2,3-dioxygenase expression was higher in large, node-positive, and ER+ tumors, while indoleamine 2,3-dioxygenase was lower in less vascularized tumors [113]. In this study, there was a greater infiltration of CD8+ and CD11b+ cells in low-indoleamine 2,3-dioxygenase tumors, whereas Treg infiltration was not associated with indoleamine 2,3-dioxygenase expression [113]. According to another study, a higher level of stromal indoleamine 2,3-dioxygenase was associated with a worse disease-free and metastasis-free survival in patients with BC [114].
Indoleamine 2,3-dioxygenase activity is commonly assessed through plasmatic kynurenine/tryptophan ratios, although they may not be a full reflection of tumor tryptophan metabolism. In patients with BC, Lyon and colleagues found a higher plasmatic kynurenine/tryptophan ratio, but there was no statistically significant difference in plasmatic levels of the two molecules [115]. It was also observed that kynurenine and tryptophan levels in plasma of BC patients were significantly lower than those in healthy controls, mainly in patients with ER- tumors and at an advanced stage of the disease [116]. As compared to ER+ cancers, Tang and colleagues observed higher plasma kynurenine levels in ER- cancers [117].
According to the above studies, indoleamine 2,3 dioxygenase 1 is elevated in tumor cells to suppress immune surveillance and promote tumor growth. This suggests that indoleamine 2,3 dioxygenase 1 might be used as a therapeutic target for the treatment of BC. Several studies have shown either an increase in indoleamine 2,3 dioxygenase 1 activity [118-120] or no difference [121,122] after receiving chemotherapy, such as paclitaxel, Mohs paste or surgery. Although limited information is available on the patient cohort, potential explanations for these different observations could be due to the pro-portion of patients with BC subtypes and/or the percentage of patients with cancers ex-pressing indoleamine 2,3 dioxygenase 1. In addition to indoleamine 2,3 dioxygenase 1, other rate-limiting enzymes have been investigated, such as tryptophan 2,3 dioxygenase and the downstream enzyme of the kynurenine pathway, kynurenine monooxygenase. There has been a positive correlation between tryptophan 2,3 dioxygenase levels and poorer overall survival, increased disease grade, and invasion/migration capability in five studies [116].
Furthermore, indoleamine 2,3 dioxygenase 1 is often detected concurrently with tryptophan 2,3 dioxygenase expression [123-127]. Additionally, tryptophan 2,3 dioxygenase expression correlated strongly with the expression of aryl hydrocarbon receptors and has been associated with enhanced tumor cell migration [123,126-128]. In the triple-negative BC subtype, kynurenine monooxygenase was found to be elevated, and it was associated with poorer survival and malignant characteristics such as node positivity. There is a high frequency of kynurenine monooxygenase amplification in invasive BC as compared to all other human cancers [129-132].
Among the small number of enzymes capable of catabolizing tryptophan, Indoleamine 2,3 dioxygenase 1 has emerged as an attractive pharmacological target due to its well-characterized structure and high affinity for tryptophan. The fact that it is the rate-limiting enzyme means that inhibition of its activity may potentially cause the whole kynurenine pathway to be blocked or reduced, and the degree of inhibition in the blood can be measured indirectly. Since most human cancers, including BC, overexpress indoleamine 2,3 dioxygenase 1, indoleamine 2,3 dioxygenase inhibitors could be used clinically in enhancing anti-tumor immunity. It has been demonstrated in in vivo and in vivo studies that combination treatment of indoleamine 2,3 dioxygenase 1 inhibition and chemotherapy drug(s) limits tumor growth. ClinicalTrials.gov has reported 97 different clinical trials examining combination therapy in cancer patients [133]. A total of six clinical trials involving patients with BC have been conducted; the different inhibitors of indoleamine 2,3 dioxygenase 1 examined were epacadostat, indoximod, and navoximod/GDC-0919 [134].”
- What is the impact of microbiota residing in BC should be discussed.
Response: Using the reviewer’s comment we modified the manuscript and now state (page 12, lines 456-471), “Mechanistically, SCFAs may play a significant role in gut-brain communication through their effects on blood–brain barrier permeability, microglial activity, neuronal function, and neuroinflammation [135], all of which have been reported following cancer treatment [136,137]. Succinate plays an active role in immunity and cancer through the stabilization of hypoxia-inducible factor (HIF)-1 [138]. Indolepropionic acid-mediated downregulation of NRF2 and upregulation of iNOS contribute to increased oxidative/nitrosative stress. An increase in oxidative/nitrosative stress reduces the stemness of cancer cells and leads to cytostasis [102]. Intestinal bacteria breakdown tyrosine into p-cresol [112] which contributes to the progression of cancer cells. The effect of lithocholic acid on BC was studied through oxidative stress and apoptosis [66,139,140]. During T regulatory cell differentiation, indoleamine 2,3-dioxygenase converts tryptophan into kynurenine, an immunosuppressive metabolite. Many types of cancer, including BC, express Indoleamine 2,3-dioxygenase [141]. A decline in bacterial diamine cadaverine concentration in the early stages of BC leads to a decrease in bacterial metabolites that are anticancer in nature [97].”

Reviewer 2 Report
In the present work, Plaza-Diaz and Álvarez-Mercado review an important topic of research, relating gut microbiota to chemotherapy-derived metabolites in breast cancer. Although it is interesting, I feel that this manuscript could be further improved if considered the following recommendations:
1) Being SCFAs one of the most studied metabolites derived from gut microbiota, why the authors say that "they are subject of a revision that goes beyond this work" instead of collecting the most relevant studies relating SCFAs and chemotherapy in breast cancer? Similarly, the role of Tryptophan metabolism by gut microbiota, especially the Kyn pathway should be also included (see: https://www.nature.com/articles/s41416-023-02245-7)
2) Notwithstanding I totally agree with the relevance of diet as a critical modulator of gut microbiota and as a coadjuvant in breast cancer patients, i feel that this section could be notably improved. For instance, exploring specific nutrients, bioactive compounds, foods or dietary patterns would bring additional value to this part.
3) Despite in line 431 it is mentioned the modulatory role of pre/probiotics and exercise, it would be necessary if possible to include a subsection collecting the main research results on these topics, as well as the main results regarding other therapeutic approaches directed to gut microbiota like fecal microbiota transplantation (FMT)
4) A figure in section 3 could aid the readers to further understand the important link between breast cancer and gut microbiota
Author Response
Mr. Farr Liu
Assistant Editor,
Thank you for providing us with the opportunity to submit a revised version of our manuscript entitled “The interplay between microbiota and chemotherapy-derived metabolites in breast cancer” to the Metabolites journal in the Special Issue titled “Effect of Diet on Gut Microbiota and Host Metabolism”. We would like to thank the reviewers for their thoughtful comments and suggestions regarding our manuscript. All comments have been considered and incorporated into the revised manuscript. Changes to the original document are tracked and highlighted in green and blue font for the reviewer's comments, with an itemized point-by-point response to the reviewers' comments.
COMMENTS FROM REVIEWER #2
The authors would like to thank the reviewer for his/her respected comments and effort made during the review process, which are highly appreciated.
Point-to-point response:
1.- In the present work, Plaza-Diaz and Álvarez-Mercado review an important topic of research, relating gut microbiota to chemotherapy-derived metabolites in breast cancer. Although it is interesting, I feel that this manuscript could be further improved if considered the following recommendations. Being SCFAs one of the most studied metabolites derived from gut microbiota, why the authors say that "they are subject of a revision that goes beyond this work" instead of collecting the most relevant studies relating SCFAs and chemotherapy in breast cancer? Similarly, the role of Tryptophan metabolism by gut microbiota, especially the Kyn pathway should be also included (see: https://www.nature.com/articles/s41416-023-02245-7)
Response: Using the reviewer’s comment, the SCFA information and tryptophan metabolism was added in the manuscript and now state, (page 10, lines 382-395), “A growing body of evidence suggests that SCFAs, particularly butyrate and propionate, can enhance chemotherapeutic agents' effectiveness by increasing tumor sensitivity or enhancing antitumor immune responses [108]. Research indicates that decreased abundances of SCFA-producing taxa (Coprococcus, Dorea, and uncultured Ruminococcus) are associated with a lower efficacy of neoadjuvant chemotherapy (cyclophosphamide, anthracycline, taxol, or herceptin) in breast cancer patients and are associated with a lower number of intratumoral CD4+ and CD8+ cells as well as peripheral CD4+ T cells [108,109].
Among survivors of BC who showed lower microbial diversity, increased Bacteroidetes, and decreased Firmicutes, the first evidence to suggest an association between the gastrointestinal microbiota composition and fear of cancer recurrence was reported [110]. As a result of a pilot study examining psychosocial factors and gastrointestinal microbiota composition in 12 BC survivors, fatigue has been associated with changes in the abundance of SCFA-producing Faecalibacterium and Prevotella as well as anxiety with changes in the abundance of Coprococcus and Bacteroides [111].body of evidence suggests that SCFAs, particularly butyrate and propionate, can enhance chemotherapeutic agents', and pages 11-12, lines 402-454,
“4.7. Tryptophan metabolites
An immunohistochemistry analysis of 203 BC samples showed that none exhibited negative staining for indoleamine 2,3-dioxygenase [113]. Indoleamine 2,3-dioxygenase expression was higher in large, node-positive, and ER+ tumors, while indoleamine 2,3-dioxygenase was lower in less vascularized tumors [113]. In this study, there was a greater infiltration of CD8+ and CD11b+ cells in low-indoleamine 2,3-dioxygenase tumors, whereas Treg infiltration was not associated with indoleamine 2,3-dioxygenase expression [113]. According to another study, a higher level of stromal indoleamine 2,3-dioxygenase was associated with a worse disease-free and metastasis-free survival in patients with BC [114].
Indoleamine 2,3-dioxygenase activity is commonly assessed through plasmatic kynurenine/tryptophan ratios, although they may not be a full reflection of tumor tryptophan metabolism. In patients with BC, Lyon and colleagues found a higher plasmatic kynurenine/tryptophan ratio, but there was no statistically significant difference in plasmatic levels of the two molecules [115]. It was also observed that kynurenine and tryptophan levels in plasma of BC patients were significantly lower than those in healthy controls, mainly in patients with ER- tumors and at an advanced stage of the disease [116]. As compared to ER+ cancers, Tang and colleagues observed higher plasma kynurenine levels in ER- cancers [117].
According to the above studies, indoleamine 2,3 dioxygenase 1 is elevated in tumor cells to suppress immune surveillance and promote tumor growth. This suggests that indoleamine 2,3 dioxygenase 1 might be used as a therapeutic target for the treatment of BC. Several studies have shown either an increase in indoleamine 2,3 dioxygenase 1 activity [118-120] or no difference [121,122] after receiving chemotherapy, such as paclitaxel, Mohs paste or surgery. Although limited information is available on the patient cohort, potential explanations for these different observations could be due to the pro-portion of patients with BC subtypes and/or the percentage of patients with cancers ex-pressing indoleamine 2,3 dioxygenase 1. In addition to indoleamine 2,3 dioxygenase 1, other rate-limiting enzymes have been investigated, such as tryptophan 2,3 dioxygenase and the downstream enzyme of the kynurenine pathway, kynurenine monooxygenase. There has been a positive correlation between tryptophan 2,3 dioxygenase levels and poorer overall survival, increased disease grade, and invasion/migration capability in five studies [116].
Furthermore, indoleamine 2,3 dioxygenase 1 is often detected concurrently with tryptophan 2,3 dioxygenase expression [123-127]. Additionally, tryptophan 2,3 dioxygenase expression correlated strongly with the expression of aryl hydrocarbon receptors and has been associated with enhanced tumor cell migration [123,126-128]. In the triple-negative BC subtype, kynurenine monooxygenase was found to be elevated, and it was associated with poorer survival and malignant characteristics such as node positivity. There is a high frequency of kynurenine monooxygenase amplification in invasive BC as compared to all other human cancers [129-132].
Among the small number of enzymes capable of catabolizing tryptophan, Indoleamine 2,3 dioxygenase 1 has emerged as an attractive pharmacological target due to its well-characterized structure and high affinity for tryptophan. The fact that it is the rate-limiting enzyme means that inhibition of its activity may potentially cause the whole kynurenine pathway to be blocked or reduced, and the degree of inhibition in the blood can be measured indirectly. Since most human cancers, including BC, overexpress indoleamine 2,3 dioxygenase 1, indoleamine 2,3 dioxygenase inhibitors could be used clinically in enhancing anti-tumor immunity. It has been demonstrated in in vivo and in vivo studies that combination treatment of indoleamine 2,3 dioxygenase 1 inhibition and chemotherapy drug(s) limits tumor growth. ClinicalTrials.gov has reported 97 different clinical trials examining combination therapy in cancer patients [133]. A total of six clinical trials involving patients with BC have been conducted; the different inhibitors of indoleamine 2,3 dioxygenase 1 examined were epacadostat, indoximod, and navoximod/GDC-0919 [134].”
2.- Notwithstanding I totally agree with the relevance of diet as a critical modulator of gut microbiota and as a coadjuvant in breast cancer patients, i feel that this section could be notably improved. For instance, exploring specific nutrients, bioactive compounds, foods or dietary patterns would bring additional value to this part.
Response: According to the reviewer's comments, we have added these new section and now the manuscript state (page 15, lines 595-623),
“5.4. Specific nutrients, bioactive compounds
BC survivors are highly likely to use dietary supplements. Diindolylmethane, a compound naturally found in cruciferous vegetables, is one of the most widely used supplements. A significant amount of experimental evidence supports the bioactivity of this bioactive compound in the prevention of BC. To test its efficacy or safety, there is a lack of well-designed human clinical trials. For a minimum of 12 months, women taking tamoxifen for primary or tertiary prevention of BC were randomly assigned to receive 150 mg diindolylmethane twice daily or a placebo. The results of this study demonstrate a favorable shift in estrogen metabolism and sex hormone binding globulin in the setting of breast cancer chemotherapy. In spite of this, the reduced levels of tamoxifen metabolites raise concerns about the potential interaction between diindolylmethane and tamoxifen, which requires further research. This data will contribute to informing the use of diindolylmethane as a dietary supplement for BC patients receiving tamoxifen, given the widespread and generally unsupported use of dietary supplements by BC survivors [175].
In a small group of 30 patients with breast cancer, the protective effect of 6 grams per kilogram of curcumine was assessed throughout the course of radiotherapy. In the interventional group, 28.6% of patients had moist desquamation, compared to 87.5% of patients in the control group [176].
Various compounds have been shown to have different levels of action regarding proliferation, apoptosis, and metastasis in breast cancer patients, mainly in vitro. Accordingly, resveratrol has demonstrated the ability to reduce negative features by acting on ER, EFGR/PI3K, and ERK1/2 pathways. For specific cases, other phytochemicals, such as lignans or curcumin, may also be beneficial in inhibiting the HER-2 pathway [177]. A new path for future research is opened by clinical evidence derived from the use of other compounds (isoflavones). In some Asian countries, consumption of this food has been associated with a lower rate of BC [177].
Furthermore, a prospective Spanish trial demonstrated that the Mediterranean diet protects against the development of BC [178,179]. It is recommended that caution be exercised when interpreting these publications, as they may incur in selection bias.”
3.- Despite in line 431 it is mentioned the modulatory role of pre/probiotics and exercise, it would be necessary if possible to include a subsection collecting the main research results on these topics, as well as the main results regarding other therapeutic approaches directed to gut microbiota like fecal microbiota transplantation (FMT).
Response: According to the reviewer's comments, we have added these new sections and now the manuscript state (pages 13-15, lines 522-593),
“5.1. Fecal microbiota transplantation
Fecal microbiota transplantation (FMT), which dates back at least 1700 years, fecal microbes are transferred from a healthy donor into the patient's intestine in an effort to restore the balance of microbiota [159]. It has been widely studied for its ability to target and modulate the human gut microbiota, which has been implicated in the treatment of numerous diseases [160]. FMT exhibits significant advantages over other treatment modalities due to its ability to maintain microbial diversity without disrupting the natural balance of the gut microbiome. Infections caused by Clostridioides difficile are commonly treated with FMT [161,162]. Currently, no studies or results have been found in Medline, EMBASE, or clinicaltrials.gov for BC and FMT.
5.2. Pre/probiotics
Gut microbiota play an important role in estrogen metabolism, which contributes to carcinogenesis. estrogens have been shown to be deconjugated by beta-glucoronidase-producing bacteria. A deconjugation of estrogen promotes its re-uptake, which may influence the risk of recurrence. Bile acid conversion is also influenced by intestinal microbes. Bile acids have similar effects to hormones and can alter metabolic pathways in distant tissues by activating receptors such as farnesoid X, which has been detected in cases of invasive BC. A murine study has demonstrated that Helicobacter hepaticus in the gut mediates the neutrophil response to breast cancer, thereby accelerating its progression [163]. As a result of chronic inflammation and epigenetic deregulation, gut bacteria can further facilitate BC development [164].
A host's gut microbiome affects the response to chemotherapy by facilitating drug efficacy or compromising anti-cancer effects and mediating toxicity. In a recent randomized trial, probiotics containing Bifidobacterium, Lactobacillus, and Enterococcus faecalis significantly reduced the risk of chemotherapy-related cognitive impairment among women receiving adjuvant chemotherapy for BC [165]. Anti-HER2 therapy is associated with specific microbiota clusters (Clostridiales, Bacteroides). In addition, there is evidence that the microbiome of the gut influences the degree of response to immune checkpoint inhibitors CTLA4 and PD-L1/PD-1 [166]. Researchers have observed that mice with specific microbiota (e.g., Akkermansia and Bifidobacterium) respond better to anti-PD-L1 therapy, and that patients with Bacteroidetes have a greater resistance to immune checkpoint inhibitor-induced colitis [164,167].
For 4 months, 34 BC survivors were randomly assigned either to follow a Medi-terranean diet plus 1 sachet of probiotics per day (Lactobacillus rhamnosus HN001, Bifidobacterium longum BB536) or to follow a Mediterranean diet alone (control group, n = 18). Compared to an Mediterranean diet alone, probiotics improve metabolic and anthropometric parameters in BC survivors with a combination of a Mediterranean diet and probiotics [168]. In other study by regulating anti-tumor immune responses, Lactobacillus acidophilus can delay the development of BC [169].
5.3. Exercise
A BC patient undergoes various treatments within one year of being diagnosed, depending on the subtype of the tumor and its stage. Symptoms associated with each treatment may negatively impact patients' health and quality of life. Exercise interventions can mitigate these symptoms when applied appropriately to patients' physical and psychological conditions [170]. Various situations and exercise routines could be related to BC, to patients who are undergoing treatment, to survivors, to patients who are receiving several medications in connection with BC. In this section, we summarize the findings of several systematic reviews.
Exercise at home and its effects on physical fitness (cardiorespiratory fitness, muscle strength, and body composition) in BC patients who are undergoing active treatment. Exercise programs conducted at home regularly are an effective strategy for improving 6-minute walk tests in BC patients undergoing active treatment. In contrast, neither muscle strength nor body composition were altered [170]. The study included 13 randomized controlled trials, involving 1569 breast cancer patients. Compared with control groups, groups who perform interventions that combine exercise and diet demonstrate significant improvements in cardiorespiratory fitness, muscle strength, body composition, quality of life, fatigue, anxiety, depression, and sleep. Conversely, exercise combined with supplementation does not result in an improvement over exercise alone or supplementation alone [171].
For cancer-related fatigue in women with BC, yoga, aerobic resistance, and aerobic yoga are recommended as inter- and post-treatment exercises to improve their physical resilience and quality of life in the long run [172].
Patients with breast cancer exhibited improved physical activity behavior for several months following exercise interventions, though the effects were small to moderate and diminished over time. After the completion of an exercise intervention, future studies should clarify how to maintain a healthy level of physical activity [173].
Oncology rehabilitation can benefit from the use of exergames, especially in BC. It is, however, necessary to conduct more rigorous studies to evaluate the effectiveness of using exergames in conjunction with conventional rehabilitation and to determine whether participants are satisfied, motivated, and adherent to the program [174].”
4.- A figure in section 3 could aid the readers to further understand the important link between breast cancer and gut microbiota
Response: Using the reviewer’s comment, a new Figure was added.

Round 2
Reviewer 2 Report
The authors have responded satisfactorily to all my queries